# DNA N^6^-Methyladenine Modification in Plant Genomes—A Glimpse into Emerging Epigenetic Code

**DOI:** 10.3390/plants9020247

**Published:** 2020-02-14

**Authors:** Jeyalakshmi Karanthamalai, Aparna Chodon, Shailja Chauhan, Gopal Pandi

**Affiliations:** Department of Plant Biotechnology, School of Biotechnology, Madurai Kamaraj University, Madurai625021, Tamil Nadu, India; jeyasweety92@gmail.com (J.K.); aparnachodon@gmail.com (A.C.); shailjachauhan982@gmail.com (S.C.)

**Keywords:** N^6^-methyladenine, DNA methylation, demethylation, epigenetic modification, 6mA detection

## Abstract

N^6^-methyladenine (6mA) is a DNA base modification at the 6th nitrogen position; recently, it has been resurfaced as a potential reversible epigenetic mark in eukaryotes. Despite its existence, 6mA was considered to be absent due to its undetectable level. However, with the new advancements in methods, considerable 6mA distribution is identified across the plant genome. Unlike 5-methylcytosine (5mC) in the gene promoter, 6mA does not have a definitive role in repression but is exposed to have divergent regulation in gene expression. Though 6mA information is less known, the available evidences suggest its function in plant development, tissue differentiation, and regulations in gene expression. The current review article emphasizes the research advances in DNA 6mA modifications, identification, available databases, analysis tools and its significance in plant development, cellular functions and future perspectives of research.

## 1. Introduction

The past few decades have witnessed several novel reports on the epigenetic modifications in DNA, RNA and histone proteins that have paved way to determine their role in various biological processes in plants and animals [1]. Among the various epigenetic alteration in nucleic acids and histones, DNA methylation plays pivotal roles to establish control in gene expression, pairing between the bases, positioning and stability of the nucleosome, replication and mismatch repair, signaling to host-pathogen interaction, transcript synthesis, X-chromosome inactivation, gene imprinting, tissue specific gene expression, virus resistance and epigenetic memory [2,3,4,5,6,7]. The occurrence of DNA methylation has been revealed in compounds such as N^6^-methyladenine (6mA), N^4^-methylcytosine (4mC), and C^5^-methylcytosine (5mC) [8,9]. The presence of 6mA, 4mC and 5mC in diverse prokaryotes and eukaryotes has been confirmed, although the degree of distribution and potential functions vary across different kingdoms [10]. The 5mC prevalence in the promoter results in definitive gene silencing [11]. The process of cytosine methylation and demethylation is well documented in plants and methylation pathway is known as transcriptional gene silencing (TGS) [12]. TGS is believed to involve the maintenance of genome integrity by inhibiting rearrangement in centromeric and telomeric repeats, suppressing transposable elements and pathogens [12,13]. Most of the studies in plants have concentrated on 5mC, while exploration of the role of 6mA has stagnated due to the unavailability of suitable methods to identify it accurately. However, recently, 6mA has been resurfaced as one of the potential reversible epigenetic mark in plants [14], which is found to be less distributed in eukaryotes when compared to prokaryotes.

Intriguingly, the occurrence of 6mA was documented in the mid-1950s in *E.coli* [15,16] by conventional techniques. 6mA modification in prokaryotes was initially reported as a part of the restriction modification system for cellular defense to distinguish the host genome from phage or plasmids via DNA methylation-sensitive restriction endonuclease [17,18]. Later studies proved its rolein DNA replication [2] and DNA repair [19]. The limitations of 6mA detection techniques, which could determine 6mA levels only if present in amounts equal to or above 0.01% of total adenine, provided compelling evidence for 6mA being undetectable in eukaryotes back in the olden days [20]. However, between1970–1981, 6mA was detected in several unicellular organisms with varying distribution (0.3–0.4%). Due to the inconsistent results, scientists reached the conclusion that 6mA was absent in multicellular organisms [21,22]. At the same time, it was assumed to be restricted to prokaryotes and unicellular eukaryotes. This notion persisted until 2015 [23,24]. Vast improvements have occurred recently in existing techniques with increased sensitivity to detect the epigenetic modifications such as high-throughput sequencing, Single Molecule Real Time Sequencing (SMRT), Immunoprecipitation (IP), Liquid Chromatography-Mass spectrometry Tandem Mass spectrometry (LC-MS/MS), enabled us to determine the distribution of base modifications along with the motifs. Studies using Ultra High-Performance Liquid Chromatography (UHPLC) revealed a noteworthy 5mC distribution in *Drosophila melanogaster* [25], resolving the existing controversy regarding the previously undetectable 5mC distribution [26,27,28]. Similar results obtained with modern efficient techniques have impelled scientists to explore 6mA in multicellular organisms, which elicited the current status of 6mA in plants. Although the role of 6mA in gene regulation has not been proven yet, experiments carried out on plants have furnished plausible evidence in favor of 6mA role in plant development, tissue differentiation [29] and mitochondrial replication [30].

## 2. Adenine Methylation and Demethylation

Active and reversible DNA adenine modification by methylation is catalyzed through the highly synchronized and directed activity of specific enzymes, generally called writer, reader and eraser proteins [31,32]. Until 1963, DNA methylation was considered to occur on unincorporated nucleotides before DNA synthesis [33,34]. Subsequent studies have demonstrated that the 6mA methylation process is modulated by methyltransferases (writers) and demethylases (erasers) in a reversible manner. Methylation can be transgenerational [35] or de novo; in either case with the involvement of specific enzymes. DNA adenine methyltransferase (DAMT) catalyzes adenine methylation solely on the single or double stranded palindromic sequence (GATC) in the presence of S-adenosyl L-methionine, a methyl group donor prevalent in DNA and RNA methylation [36,37].

Though DNA adenine methyltransferase (DAM) in bacteria, DAMT-1 in *C. elegans* [26], METTL4 in mammals [38] and *Bombyx mori* [39], N6AMT1 in humans [40] are known writer proteins, little is known about methyltransferases in higher plants [41]. Nevertheless, the activity of a Mg^2+^ or Ca^2+^- dependent wheat N6-adenine DNA-methyltransferase (*Wadmtase*) purified from aging wheat coleoptiles, subsequently revealed its potential role in 6mA methylation. *Wadmtase* has the capability to recognize a hexanucleotide sequence (TGATCA) but not a tetranucleotide sequence (GATC) to methylate adenine [30]. Further, it has been shown that the tRNA methylase 11 (TRM11) family of RNA methylases act as DNA adenine methylases in plant mitochondria, which is still uncertain since it lacks a mitochondrial DNA targeting peptide that is essential for mitochondrial genome methylation [31]. Yet another protein, the MT-A70 domain family of enzymes (IME4-like family) thought to be evolved from MunI-like enzymes of prokaryotes is widely distributed and highly conserved in eukaryotes. These observations suggest its possible role as a methyltransferase in higher organisms [42]. Structurally, MT-A70 domain family enzymes possess a 7-β-strand methyltransferase domain at their C-terminus fused to a predicted α-helical domain at their N-terminus [31].

In plants, cytosine demethylation is mediated by DNA glycosylases via a base excision repair (BER) mechanism [43]. Like cytosine demethylation, adenine demethylation is known to proceed in either of two ways—via hydroxymethyladenine or hypoxanthine [44] (Figure 1). The hypoxanthine involved pathway is mediated by 6mA deaminase, which is further probably modulated by the BER pathway [45]. The α-ketoglutarate-dependent hydroxylase (ALKB) family of enzymes known as dealkylating agentsare prevalent and conserved ranging from prokaryotes to higher eukaryotes [44]. The murine ALKBH4, was found as an orthologue of DNA 6mA eraser in *D. melanogaster* and *C. elegans* [38] and also could act as a complement protein with N6AMT1 to control 6mA levels in genomic distribution in different cell types [40]. In *Oryza sativa,* OsALKBH1 is the only protein found in the nucleus and cytoplasm, which will potentially mediate demethylation of adenine [46]. OsALKBH1, which belongs to the above family, has a preserved domain homology with iron interaction and a 2-oxoglutarate Fe (II)-dependent dioxygenase binding domain (2OGFeDOs). Similar domains and motifs have been documented in *Arabidopsis thaliana* ALKBH1, which possesses the capacity to oxidatively reverse 6mA in vitro and in vivo [47]. The demethylation activity of ALKB family of enzymes are examined to process via hydroxymethyladenine, which further releases the aldehyde group and apparently gets converted back into the adenine base [44]. Interestingly, these ALKB members are also involved in the catalysis of oxidative demethylation of RNA [48].

Apart from the writer and eraser proteins, reader proteins are the major category of proteins involved in determining the role of methylation [49]. These specific effector molecules capable of recognizing the 6mA signal allow the various impacts of 6mA including gene expression and regulation. In prokaryotes, reader proteins like SeqA are capable of recognizing hemimethylated GATC sites and binding on them [50,51], thus preventing premature DNA replication before cell division [52,53]. The Jumu protein belonging to the Fox-family protein present in *D. melanogaster* is found to recognize the 6mA mark on *Zelda* and regulate the maternal-to-zygotic transition by regulating *Zelda* expression [54]. Despite this fact, 6mA readers in plants remain elusive. However, ECT2, an RNA N^6^ -methyladenosine (m6A) reader protein, that binds with m6A and promotes proper trichome development in *A. thaliana* has been identified [55]. Apart from this, based on the studies with H3K4me2 spr-5 mutant *C. elegans,* a correlation between histone methylation and adenine methylation has also been reported [26] implying an interplay between the histone and DNA methylation. Despite this fact, understanding reader proteins in eukaryotes remains largely unknown and absolute focus has to be given to decipher this further. 

## 3. Abundance of 6mA Level in Plants and Possible Functional Importance

Until recently, the 6mA abundance and distribution has been investigated in a small number of plants [30,46,56,57], unicellular algae [58] and multilineage fungi [59]. The 6mA levels show considerable variation in prevalence and effect on gene expression in different species owing to their different distribution patterns. In general, it can be stated that the 6mA level in prokaryotes is higher (0.002–2.7%) [16] when compared with eukaryotes (0.000006–0.8%) [60,61]. The reasons for the low distribution in eukaryotes are unknown, but probably it can be linked with the difference in the size, genome complexity, presence and length of palindromic sequences and the complicated methylation processes.

*C. reinhardtii*, where 6mA frequency is observed in nearly 14,000 genes (84% of total genes), is reported to have a bimodal distribution with a distinct pattern of high 6mA level around transcription start sites (TSSs). Transcriptome sequencing data provide evidence for active gene expression when 6mA is present around TSS [58]. The variation in the width of peaks observed in 6mA-IP-sequencing data suggests the presence of multiple motifs, and a major portion of 6mAlies in palindromic AT dinucleotides while GATC and CATG also marks methylation. Intriguingly, a similar observation is also reported in *Volvox carteri* at GATC sites (Table 1) [62]. In contrast, 6mA profiles in *A. thaliana* have revealed that 32% of the total 6mA is distributed inside gene bodies with significant deposition on exons and transposable elements [56]. Further mining on the MEME-ChIP data showed two novel 6mA motifs, ANYGA and ACCT, apart from GAGG and AAAGAV. Like *C*. *reinhardtii* [58] and early-diverging fungal lineages [63], 6mA is enriched around the TSS in *A.thaliana* and correlated with active gene expression [56], suggesting the possibility that 6mA may preserve a definitive role in the course of evolution. Although 6mA showed a distinct pattern of increased methylation near TSS and internal decline at TSS in transposable element genes, the same was not in total agreement in the case of protein coding genes in *A. thaliana*. Besides, the degree of 6mA seems to diverge in different developmental stages and tissues [56].

The 6mA methylome profiles in *O. sativa* was recently deciphered [46] where 0.2% of all the adenine was marked with 6mA with an accumulation on nearly 25% of coding genes, 36% of transposable element-related genes and 14% of transposable elements. Although this level of abundance was comparable to that of green algae, the identified motifs, AGG and GAGG, were non-palindromic, unlike the latter. The spatially distinct pattern of 6mA distribution revealed higher level of occurrence in promoter and intergenic regions than gene bodies. 6mA occurrence depletes from TSS with dramatic rise at the transcriptional termination sites, signifying its negative role in gene expression. As like 5mC, 6mAin promoter is associated with reduced gene expression while the same on gene bodies represents active transcription in rice plants [46]. Combining these observations, it can be proposed that 6mA favors to limit the gene expression in rice plants.

Though 6mA distribution and specific occurrence is naive in wheat and maize plants, potential motifs of TGATCA capable of getting methylated by *Wadmtase* and GATC have been identified in each respectively [30,64]. Similarly, GATC motif has been identified inthe barley promoter region, which showed increased expression of a transfected plasmid [65]. In the Rosaceae family, two of the plant species, *Fragaria vesca* and *Rosa chinensis,* are subjected to methylome profiles, which have revealed ADSYA and ADGYA as consensus motifs respectively [57]. Besides, studies involving introduction of exogenous *DAM* gene from bacteria helped to identify methylation site (GATC) in transgenic tobacco, wheat and tomato in association with active transcription from the modified promoters resulting in phenotypical changes [30,66]. Overall, in a few species, 6mAis abundant around TSS, which could positively correlate with gene transcription [56,58,63]. However, these phenomena are found to be absent in rice [46], suggesting that 6mA may harbor diverse functions in different plants.Most of the features like prevalence, distribution and possible function of 6mA in plants are yet to be determined.

## 4. 6mA Detection Methods

The 6mA level is extremely low in eukaryotes, especially in plants, which renders 6mA detection by conventional methods complicated and cumbersome. However, despite its reduced levels, the periodicity in distribution and controlled deposition implicates its possible crucial roles in genome topology [58]. With the advent of recent sophisticated techniques with high sensitivity and precision (Table 2), 6mA distribution pattern, motifs, enzyme machinery and its potential roles have been revealed in eukaryotes [41,58].

## 5. 6mA Methylome Databases

Different databases, which provide a single platform to avail the 6mA methylation profiles of diverse species have been developed. These databases accumulate information from existing genomic databases and publicly available SMRT sequencing data, and allow comparative studies on methylation in various species. 

MethSMRT is the first resource for DNA 6mA and 4mC methylomes, which are generated from the publicly available SMRT sequencing data [79]. This will allow genome browsers to query and envisage methylation-related profiles like single nucleotide polymorphisms (SNPs), and gene annotation. It also analyzes and downloads 6mA and 4mC methylomes for prokaryotes and eukaryotes. In addition to methylation profiles, 6mA and 4mC methylome statistics can be performed along with prediction methylation motifs for individual species, which enables them comparing methylation in different species. 

DNA modification Database for Rosaceae (MDR) is the first source for displaying and storing DNA 6mA and 4mC methylomes from SMRT sequencing data of Rosaceae plants. The present version of MDR includes datasets of Rosaceae plants including *F. vesca* and *R. chinensis*. MDR imparts a platform to retrieve the metadata from the Genomic Database for Rosaceae (GDR) [80] and other sources to identify the methylation modification. It further provides the user with information about possible methylation motifs, gene annotation, and gene ontology [57].

DNAmod is an open-source DNA modification database) that catalogues DNA modifications and provides a single resource to launch an investigation of their properties using Chemical Entities of Biological Interest (ChEBI) database [81]. This also offers reference to modified base nomenclature and offers the potential to track current progresses within the field[82].

## 6. Bioinformatic Analysis Tools for 6mA

After the detection of 6mAin eukaryotes with the modern sophisticated techniques, the subsequent obstacle is analysis of 6mA methylome data and identification of the methylation motifs.In recent times, researchers have succeeded in developing bioinformatic tools for the detection and prediction of the potential methylation sites with high specificity and accuracy. Although most of the tools described below are proficient to work for rice genome, some of them have been found to be applicable to the other plants as well:(1)SDM6A: A sequence-based two-layer ensemble approach for the effective prediction of 6mA novel putative sites and non-6mA sites in the rice genome using Integrative Machine-Learning Framework [83].(2)SNNRice 6mA: Another 6mA identification method for the rice genome using a simple and lightweight deep learning model and the evaluation based on five metrics like accuracy, sensitivity, specificity, Matthews correlation coefficient (MCC), and area under the curve (AUC)) [84].(3)i6mA-DCNP: Identification and prediction of 6mA sites in rice genome with high quality computational model. The prediction is based on encoding the genomic DNA samples using dinucleotide composition and optimized dinucleotide-based DNA properties [85].(4)csDMA: Identification and prediction of DNA 6mA modification in different species via Chou’s 5-step rule using three encoding features and different algorithms to generate the feature matrix [86].(5)iDNA6mA-Rice: Evaluation ofm6A sites in the rice genome using the machine learning random forest algorithm to formulate the sample as an input to discriminate from the methylated and non-methylated sites [87].(6)iDNA6mA-PseKNC: A sequence based predictor that enables identification of DNA 6mA sites with 100% specificity and 96% accuracy without going for complex mathematical formulae [88].(7)iDNA 6mA: Identification of 6mA sites in the rice genome using deep learning method based on conventional neural network, which takes single input of DNA sequences [89].(8)i6mA-Pred: Identification of 6mA sites in the rice genome with 83% accuracy in which the DNA sequences were formulated and encoded effectively by the use of chemical property and frequency of nucleotide based on support vector machine method [90].(9)MM-6mA-Pred: Identification of 6mA and non-6mA sites by significant difference in transition probability among adjacent nucleotides based on Morkov Model with better prediction than i6mA-Pred) [91].(10)DEEP6mA: A superior performance tool allowing identification of 6mA sites in plants with an overall prediction accuracy of 94% using convolutional neural network (CNN) to extract high-level features in the sequence and a bi-directional long short-term memory network (BLSTM) to learn dependence structure along the sequence [92].(11)FastFeatGen: This tool predicts the 6mA sites in the genome using machine learning approach with the motif features. The advantage of this tool is its speed due to the multi-threading and shared memory process [93].

## 7. Conclusions and Future Aspects

Currently, unraveling the epigenetic mechanisms beyond the existing genetic code is indispensable to understand how living organisms are able to cope up with their varying environment. DNA 6mA, initially identified as a defense mechanism to distinguish host DNA from alien ones, is a predominant modification in prokaryotes whereas in eukaryotes 5mC is a widespread DNA modification that is associated with gene silencing. 6mA, though discovered in the mid-1950s and studied well in prokaryotes, had not been considered significant in higher organisms until the introduction of enhanced analytical techniques that have allowed its detection and quantification. Presently, 6mA abundance, motifs, distribution, methylation and demethylation enzymes and its potential roles have been deciphered in different species and found to be variable in each species. Since the whole genome sequence of many eukaryotes remains unavailable, it is quite difficult to analyze 6mA levels and its epigenetic modulation in many species. The ability of *DAMT* to preferentially methylate TGATCA or AAAGAV adds to the possibility of higher plants having writers capable of recognizing hexamers or even longer motifs rather than dimers, trimers or tetramers, which needs to be envisaged further. Besides genome complexity, size and reciprocal process between writer and eraser, the lower probability of such motif repetition could justify the abysmal level of 6mA in eukaryotes than in prokaryotes. Contrasting patterns in gene expression observed in 6mA methylation studies in different organisms could be due to the influence of 5mC already present in them that need to be deciphered in detail. Another possibility is that both 5mC and 6mA abundance is indispensable for the effective silencing of a particular gene, which also needs to be explored. In *A. thaliana*, 6mAoccurrence at TSS leads to active gene transcription as observed in *C. reinhardtii* and early-diverging fungal lineages. Though the exact reasons for transcriptional activation is not known, it is possible that 6mA at TSS probably signals to recruit the reader protein, which in turn mobilizes the demethylase for the preferential erasing of 5mC and 6mA to relieve the repression in the promoter region. Therefore, further investigation is needed to portray the unambiguous role for 6mA in the presence or absence of 5mC. Many features of 6mA are conserved and divergent in distribution pattern and how the fundamental functions in epigenetic modifications have been evolved needs to be analyzed genetically as well as by biochemical characterization of writers, readers and erasers. Since the role of 5mC has been proven in tissue differentiation, the distribution level of 6mA also vary from one cell to another cell of an organism in coding genes at constant and variable region. So, unlike the known genetic code, epigenetic code, which is cell-specific rather than organism specific, is yet to be fully comprehended. Transgenerational studies about 6mA epigenetic memory inheritance could potentially reveal the role of more proteins involved in the methylation machinery as 6mA could either be directly inherited over generations or can coordinate similar epigenetic marks to regulate the 6mA level in subsequent generations. 

## Figures and Tables

**Figure 1 plants-09-00247-f001:**
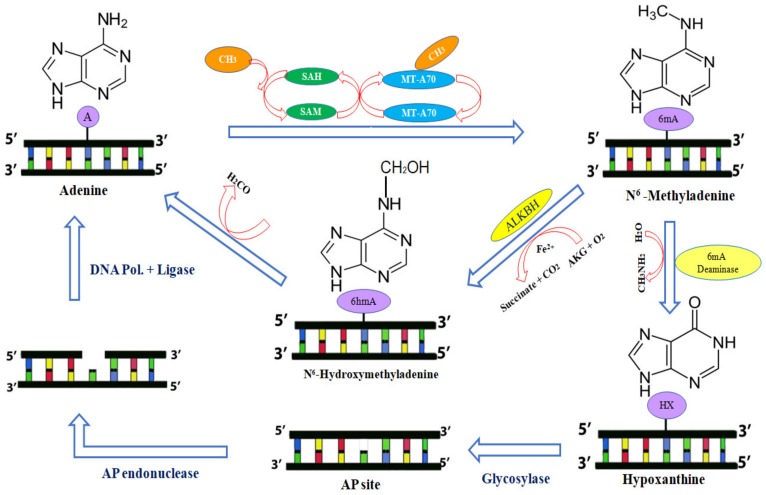
Mechanism of reversible DNA Adenine methylation (6mA)—Using S-adenosyl L-methionine as a methyl group donor, the writer MT-A70 protein family adds methyl group in the 6^th^ position of purine ring of adenine. It has been shown that 6mA can be converted to unmethylated adenine by the two mechanisms. In the first pathway, 6mA reverts back to adenine by ALKBH family proteins (eraser) by oxidative demethylation via intermediary hydroxymethyladenine (HO-6mA). In yet another mechanism, 6mA is converted back to adenine by 6mA deaminase (eraser) releasing the intermediate product of hypoxanthine, which is recognized and cleaved by glycosylase leaving an apurinic site (AP site). This AP site is further rectified by base excision repair involving AP endonuclease, DNA polymerase and DNA ligase. The 6mA reader proteins, which are efficient enough to recognize methylated adenine motifs, remain unknown.

**Table 1 plants-09-00247-t001:** The distinctive attributes of 6mA in plants compared with *E.coli*and human.

Organism	Motif	Pattern of Distribution	Potential Role	Abundance% (6mA/A)
*Escherichia coli* [16]	GATC, TGAA	Complete genome	DNA repair, replication & defense	1.7–2.4
*Homo sapiens* [40]	[G/C]AGG[C/T]	Complete genome	Gene transcriptional activation	0.051
*Chlamydomonas reinhardtii* [58]	GATC, CATG	TSS	Gene expression	~0.4
*Volvox carteri* [62]	GATC	TSS	Unknown	0.3
*Triticum aestivum* [30]	TGATCA	Mitochondrial DNA	Mitochondrial DNA replication	-
*Zea mays* [64]	GATC	Gene body	DNA repair	0.0015
*Arabidopsis thaliana* [29]	ANYGA, GAGG, ACCT, AAAGAV	Exon, TSS	Stress response, gene activation	~0.2
*Oryza sativa* [46]	GAGG, AGG	Promoter and Intergenic region	Gene repression	0.2
Rosaceae [57]	ADSYA, ADGYA	Gene body	Photosynthesis	-
Transgenic *Nicotiana tabacum* [67]	GATC	-	Gene regulation	-
Transgenic *Solanum lycopersicum* [68]	GATC	-	Tissue differentiation	-

N-Any nucleotide, Y-Pyrimidine (C or T), S-Strong (G or C), D-Not C (A or G or T), V-Any nucleotide except T.

**Table 2 plants-09-00247-t002:** Various 6mA detection methods and their significance.

Method	Significance
SMRT—A Third Generation Sequencing Method	Based on enzyme kinetics, 6mA bases are identified by highly sensitive inter-pulse duration ratios in the sequencing data [69].
MeDIP-seq(Methylated DNA immunoprecipitation sequencing)	The 6mAin DNA is discovered by using specific antibody by applying high throughput DNA sequencing [70].
Dam ID (DNA adenine methyltransferase Identification)	Proteins of interests are fused with Dam and the binding sites are identified by restriction digestion and mapping [71].
Dam IP (DNA adenine methyltransferase Immunoprecipitation)	Combination of a mutant form of DAMT with 6mA antibody to recognize N-6-methylated DNA followed by IP based enrichment and detection by qRT-PCR with sequence specific primers [72].
UHPLC-TQMS (Ultra High-Performance Liquid Chromatography-Triple Quadrupole Mass Spectrometry)	Sensitive detection and quantification of 6mA by high performance liquid chromatography coupled with mass spectrometry [73]
MEME-ChIP (Multiple EM for motif elicitation-Chromatin immunoprecipitation)	Identification and analysis of 6mA motif from the large set of sequence, which were obtained from ChIP [74].
6mA CLIP Exo (Immunoprecipitation, Photo Crosslinking-Exonuclease Digestion).	Detection of 6mA motifs in genomic DNA fragments incubated with 6mA antibodies followed by UV cross-linking, immunoprecipitation and exonuclease digestion [58].
6mA-RE(Restriction Endonuclease).	Restriction digestion of unmethylated 6mA motif with *Cvi*AII (CATG) or *Dpn*II (GATC) leaves methylated fragments which can be amplified by PCR [58].
Metal ion mediated replication and Rolling Circle Amplification.	This method is able to differentiate the methylated and non-methylated sequences. The non-methylated sequences are ligated and circularized with Ag^2+^ padlock probes where mismatches are stabilized by metal ions and primer extension processed by polymerases [75].
Dot blot	Detection of 6mA abundance using specific antibodies and distinction of changes in 6mA level in different tissue or same tissue at different time point [46,76].
MCSeEd (Methylation content sensitive enzyme double-digest restriction-site-associated DNA (ddRAD) technique)	Identification of 6mA motif using parallel restrictions carried out by combinations of methylation insensitive and sensitive endonucleases, followed by next-generation sequencing [77].
Nanopore	Identification of DNA methylation at specific genomic position by Nanopore signals where it shows the electric spikes [78].

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
