# Peer review of "DNA N6-Methyladenine Modification in Plant Genomes—A Glimpse into Emerging Epigenetic Code"

_plants, 2020, doi:10.3390/plants9020247_

Round 1

Reviewer 1 Report

Karanthamalai et al., opted for an intriguing topic related to epigenetics and summarized the latest findings on the DNA N6-methyadenine (6mA) modification as a potential epigenetic mark in eukaryotes. As the authors described in the submitted review article, the 6mA was discovered several decades ago. Hence, review articles have been available so far. In this regard, the author’s review article is not entirely unique or new. Nevertheless, I find the submitted review manuscript worth a publication due to the fact that available databases and tools for the analysis (including prediction) for 6mA are provided there. Such technical information, along with its historical point of view and the updates on the relevance of 6mA to gene regulation as a potential epigenetic mark in plants, makes the author’s article a nice package as a whole that is likely to attract readers. Overall, their manuscript was adequately written in grammar and spelling with a logical flow, however, I would advise authors to consider the following points. Figure 1 and the “Conclusion and Future aspects” require moderate changes. I sincerely hope the suggestions described below would help to improve the quality of the manuscript. 

(1) Whenever seeing “6mA methylation” throughout, I was always perplexed with it. Does it mean “additional” methylation on N6-methyadenine? N6-adenine methylation by mistake? Or N6-methyadenine? Please make sure if “6mA methylation” makes sense to you indeed.

(2) Please add a comma before “which” throughout. Alternatively, change to “that” if appropriate.

(3) Please be consistent with the names of the organism as “ thaliana” appears in one place and “Arabidopsis” emerges in the others…

(4) “6mA presence” appears countless times. Do you really need the word “presence”???

(5) Line number 45-47: Although the authors briefly mentioned the 6mA modification in E. coli serving as cellular defense mechanism, it would be helpful if authors include a couple of sentences stating that 6mA in prokaryotes is prevailing modification that was initially reported as a part of restriction-modification system to distinguish their genome from phage or plasmids via DNA methylation-sensitive restriction endonuclease. Reference: Arber, W.; Dussoix, D. Host specificity of DNA produced by Escherichia coli: I. Host controlled modification of bacteriophage λ. J. Mol. Biol. 1962, 5, 18–36.

(6) Figure 1 requires some cosmetic make-ups. Is it possible to have a consistent font (the same as text font)? After the rotation of some words (such as H2CO or Succinate) they are deformed. The name of the enzyme, 6mA Deaminase, which is black in green background is difficult to recognize. To be consistent with the other names in colored background, changing to the white font color is advisable. To be consistent with structures of adenine, N6-methyladenine, and N6-hydroxymethyladenine, the structure of Hypoxanthine requires horizontal flipping. Or is there a particular reason for Hypoxanthine to have such orientation? The title of Figure 1 sounds odd to me because Figure 1 illustrates not only methylation but also demethylation. I provided an alternative title for Figure 1 in an attached pdf.

(7) Similar to Table 2, please include references in Table 1 if available.

(8) Line 62-64: Although the role of 6mA in plant regulation has not been proven yet, but experiments carried out on plants have furnished strong evidence in favor of 6mA role in plant development, tissue differentiation [25] and mitochondrial replication [26].

I am not entirely sure that article 25 provides “strong” evidence for the role of 6mA in plant development and tissue differentiation because Liang et al., in reference 25 found a positive correlation between more frequent 6mA deposition near TSS and active gene expression over vegetative-to-reproductive phase transition in Arabidopsis areal part. Substituting “strong” to a more relevant word is preferred. “plant regulation” is vague. As for referring to the role of 6mA in mitochondrial replication, citing [35], in place of citing [26], is a more suitable one, I think.

(9) Line 66-67. “6mA methylation occurs by the highly synchronized and directed activity of specific enzymes, generally called writer, reader and eraser proteins [27,28].

This sentence describes reversible 6mA modification through writer, reader and eraser proteins. The verb, occur, does not look like an appropriate choice. Did you mean “Active and reversible DNA adenine modification by methylation is catalyzed through highly synchronized and….”?

(10) In Table 2, I would suggest MeDIP-seq (as reference 58 says), instead of MeDIP itself. If agreed, please phrase its significance accordingly.

(11) Line 188: Please check the font size of the URL. It looks bigger. The same applies to URLs in lines 200 and 201

(12) Line 268-270: “In Arabidopsis, 6mA presence at transcription start site led active gene transcription, possibly that 6mA at TSS might lead to recruiting demethylase to erase the 5mC and 6mA in the promoter region.

This sentence is hard to make out. Do you mean that 6mA at TSS might diminish not only 5mC but also its own 6mA methylations for active transcription??? I understand that this is where as many alternative interpretations are raised as possible. I am also aware that it is a big challenge to come to the solid conclusion that the DNA methylations are the cause or the consequence of gene regulation but please elaborate it.

(13) Line 272: The sentence “Many features of 6mA methylation are divergent” is ambiguous to me.

(14) Minor correction can be seen in an attached pdf document.

(15) At last, I suggest authors to elaborate “Conclusion and Future aspects” Good start would be, as authors described in line 274, to concisely mention the requirement of epigenetic code beyond the genetic code, as a part of mechanisms of living organisms to cope with the varying environment. DNA 6mA, initially found as a defense mechanism to distinguish host DNA from the alien ones, is a prevalent modification in prokaryotes whereas in eukaryotes 5mC is dominant DNA modification that is associated with gene silencing. Due to its poor abundance in eukaryote genomes and the unavailability of sensitive and robust methods for detection, 6mA remained insignificant in higher organisms until recently... (Continue line 255)…

My suggestion to authors carries on by stressing the urge to discover the DNA 6mA reader and its relevance to epigenetics. In my point of view, the drawback in the 6mA research is the lack of genetic evidence. As far as I am concerned, at least one RNA 6mA reader protein, ECT2, was found in Arabidopsis and loss-of-function mutant phenotypes, as well as biochemical properties, was documented (Wei et al, Plant Cell 2018). Similarly, in conjunction with histone acetylation, epigenetic relevance of DNA 6mA in C. elegans was corroborated in nmad-1, a mutant defective in N6-methyladenine demethylase 1 (Greer et al, 2015 Cell).    

Line 274-275: Authors claimed that the epigenetic code, which is cell-specific rather than organism-specific, is yet to be fully comprehended.

Agreed yet I feel like elaborating this, too by giving the above-mentioned C. elegans work as an example, telling that DNA 6mA is detectable in most cells by two independent antibodies raised against DNA 6mA (Greer et al, 2015 Cell). 

Author Response

We are extremely thankful to you for your critical comments and advices on our submitted manuscript. We would like to let you know that we have rectified the errors that came from our part and incorporated your suggestions. The modifications in the manuscript have been made with track changes.

(1) Whenever seeing “6mA methylation” throughout, I was always perplexed with it. Does it mean “additional” methylation on N methyadenine? N -adenine methylation by mistake? Or N methyadenine? Please make sure if “6mA methylation” makes sense to you indeed.

 We are sorry for the inadvertent mistake from our part. We had intended to mean methylation on N6- adenine. The mistake has been rectified by deleting the word “methylation” throughout the draft.

(2) Please add a comma before “which” throughout. Alternatively, change to “that” if appropriate.

 Kindly note this has been corrected throughout the text.

(3) Please be consistent with the names of the organism as “ thaliana” appears in one place and “Arabidopsis” emerges in the others…

 It is a mistake from our side, changed accordingly in the text.

(4) “6mA presence” appears countless times. Do you really need the word “presence”???

 As per your suggestion, either deleted or changed into occurrence or prevalence wherever it is necessary.

(5) Line number 45-47: Although the authors briefly mentioned the 6mA modification in E. coli serving as cellular defense mechanism, it would be helpful if authors include a couple of sentences stating that 6mA in prokaryotes is prevailing modification that was initially reported as a part of restriction modification system to distinguish their genome from phage or plasmids via DNA methylation-sensitive restriction endonuclease. Reference: Arber, W.; Dussoix, D. Host specificity of DNA produced by Escherichia coli: I. Host controlled modification of bacteriophage λ. J. Mol. Biol. 1962, 5, 18–36.

 Please find that we have modified the sentence as per your suggestion along with the given reference.

(6) Figure 1 requires some cosmetic make-ups. Is it possible to have a consistent font (the same as text font)? After the rotation of some words (such as H CO or Succinate) they are deformed. The name of the enzyme, 6mA Deaminase, which is black in green background is difficult to recognize. To be consistent with the other names in colored background, changing to the white font color is advisable. To be consistent with structures of adenine, N -methyladenine, and N -hydroxymethyladenine, the structure of Hypoxanthine requires horizontal flipping. Or is there a particular reason for Hypoxanthine to have such orientation? The title of Figure 1 sounds odd to me because Figure 1 illustrates not only methylation but also demethylation. I provided an alternative title for Figure 1 in an attached pdf.

 Thank you for your advises on improvising the figure. All your suggestions have been taken into account and the picture and its title has been modified accordingly.

(7) Similar to Table 2, please include references in Table 1 if available.

Please note that the references have been provided in the table.

(8)  Line 62-64: Although the role of 6mA in plant regulation has not been proven yet, but experiments carried out on plants have furnished strong evidence in favor of 6mA role in plant development, tissue differentiation [25] and mitochondrial replication [26].

I am not entirely sure that article 25 provides “strong” evidence for the role of 6mA in plant development and tissue differentiation because Liang et al., in reference 25 found a positive correlation between more frequent 6mA deposition near TSS and active gene expression over vegetative-to-reproductive phase transition in Arabidopsis areal part. Substituting “strong” to a more relevant word is preferred. “plant regulation” is vague. As for referring to the role of 6mA in mitochondrial replication, citing [35], in place of citing [26], is a more suitable one, I think.

We have replaced the word “strong” with “plausible “in line 67 and “plant regulation” with “gene regulation” in line 66. We have also changed the reference in line 68 according to your suggestion.           

(9)  Line 66-67. “6mA methylation occurs by the highly synchronized and directed activity of specific enzymes, generally called writer, reader and eraser proteins [27,28].”

This sentence describes reversible 6mA modification through writer, reader and eraser proteins. The verb, occur, does not look like an appropriate choice. Did you mean “Active and reversible DNA adenine modification by methylation is catalyzed through highly synchronized and….”?

 We accept your suggestion and have modified the sentence as per your comment.              

(10)  In Table 2, I would suggest MeDIP-seq (as reference 58 says), instead of MeDIP itself. If agreed, please phrase its significance accordingly.

 We totally agree with your point. The correction has been made accordingly.

(11)  Line 188: Please check the font size of the URL. It looks bigger. The same applies to URLs in lines 200 and 201.

 Kindly note this has been changed.   

(12)     Line 268-270: “In Arabidopsis, 6mA presence at transcription start site led active gene transcription, possibly that 6mA at TSS might lead to recruiting demethylase to erase the 5mC and 6mA in the promoter region.”

This sentence is hard to make out. Do you mean that 6mA at TSS might diminish not only 5mC but also its own 6mA methylations for active transcription??? I understand that this is where as many alternative interpretations are raised as possible. I am also aware that it is a big challenge to come to the solid conclusion that the DNA methylations are the cause or the consequence of gene regulation but please elaborate it.

Here, we were trying to deduce the possible role of 6mA in gene regulation. The sentence has been modified and elaborated. Hope that the point is clear now.           

(13)   Line 272: The sentence “Many features of 6mA methylation are divergent” is ambiguous to me.   

This also has been modified to give more clarity to the point.                      

(14)  Minor correction can be seen in an attached pdf document.

 Thank you for your kind effort. All the changes have been incorporated in our manuscript.

(15)  At last, I suggest authors to elaborate “Conclusion and Future aspects” Good start would be, as authors described in line 274, to concisely mention the requirement of epigenetic code beyond the genetic code, as a part of mechanisms of living organisms to cope with the varying environment. DNA 6mA, initially found as a defense mechanism to distinguish host DNA from the alien ones, is a prevalent modification in prokaryotes whereas in eukaryotes 5mC is dominant DNA modification that is associated with gene silencing. Due to its poor abundance in eukaryote genomes and the unavailability of sensitive and robust methods for detection, 6mA remained insignificant in higher organisms until recently... (Continue line 255)…

My suggestion to authors carries on by stressing the urge to discover the DNA 6mA reader and its relevance to epigenetics. In my point of view, the drawback in the 6mA research is the lack of genetic evidence. As far as I am concerned, at least one RNA 6mA reader protein, ECT2, was found in Arabidopsis and loss-offunction mutant phenotypes, as well as biochemical properties, was documented (Wei et al, Plant Cell 2018). Similarly, in conjunction with histone acetylation, epigenetic relevance of DNA 6mA in C. elegans was corroborated in nmad-1, a mutant defective in N6-methyladenine demethylase 1 (Greer et al, 2015 Cell).   

Line 274-275: Authors claimed that the epigenetic code, which is cell-specific rather than organism-specific, is yet to be fully comprehended.

Agreed yet I feel like elaborating this, too by giving the abovementioned C. elegans work as an example, telling that DNA 6mA is detectable in most cells by two independent antibodies raised against DNA 6mA (Greer et al, 2015 Cell).

 We have included all the above mentioned references in the manuscript and modified the “Conclusion and future aspects” part accordingly. Wei et al, Plant Cell 2018 has been cited in line 133 and Greer et al, 2015 Cell has been included in line 136.

Reviewer 2 Report

The authors reviewed the identification of 6mA, databases, analyzing tools, and potential functions in plant development. Overall it described the progress in the field and the weakness is that it lacks insights into the field. 

A few comments:

Please use words carefully and make sure they accurately describe the contents. For example, Line 17 "naive" is not an appropriate word and need to replaced; Line 17 "indispensable" is too strong word since there is no strong evidence to demonstrate that 6mA is indispensable, even for 5mC, I rarely see people use "indispensable" to describe its function. Line 95, Is "ALKBH1" a rice protein or Arabidopsis protein? If it refers to ALKBH1 in rice, it shall write as OsALKBH1 to be consistent with OsALKBH1 on Line 96. Please be consistent with format of subtitles: For example, to be consistent of the subtitle 2, the subtitle 3 should be: Abundance of 6mA Level in Plants and Possible Functional Importance (Only "P" in "Plants" was in upper case in the original text). Subtitles 4, 5, 6 have the same issue.  Delete ":" at the end of Subtitles 2, 3, 4, 5.

Author Response

We thank you for your valuable remarks on the submitted manuscript. We absolutely agree with your viewpoints and have modified the manuscript as per your suggestions. Kindly note that the following changes have been made in the manuscript based on your points.

The word “naive” in line 17 has been replaced with “less known”. The word “indispensable” in line 17 has been deleted. In line 104, it refers to ALKBH1 in rice and it has been changed to OsALKBH1. The format of the subtitles has also been made uniform throughout the text.

Reviewer 3 Report

In this review, the authors discussed the recent advances in DNA 6mA modification and available database and analytical tools. In general, the authors did a good job, especially the databases and analytical tools.

6mA studies are relatively more advanced in animals than plants right now. The authors need to incorporate more recent publications in this review. Some of the citations are not current, for example page 2 citations 34 and 36. Mammalian Mettle4 is reported to catalyze 6mA deposition and Alkbh4 erases it (Soo-Mi Kweon et al, Molecular cell, 2019).

The functions of several key genes involved in 6mA have reported in animals. For instance, fox-family protein Jumu is reported to bind 6mA-marked DNA and acts as a maternal factor to regulate the maternal-to-zygotic transition in Drosophila (He et al, Nature comm, 2019). And, in Drosophila, 6mA is reported to associate with tissue-specific expression of developmental genes (Shah et al, G3, 2019). The authors need to discuss whether their counterparts have been discovered in plants. Since “the role of 6mA in plant regulation has not been proven yet” (stated in Page 2 line 62), this may shed light on potential functions of 6mA in plants.

For table 1, instead of using E coli, I think it would be more informative if the distribution of 6mA in human genome is added (Xiao et al, Molecular cell, 2018).

Author Response

We are grateful to you for your valuable comments and suggestions for our manuscript. We have tried to incorporate them and modify the manuscript accordingly. The following changes have been made to the text based on your valid comments.

Recent reference (Soo-Mi Kweon et al, Molecular cell, 2019) has been added in the manuscript in lines 81 and 101 according to your suggestion. He et al, Nature Communication, 2019 has been added in lines 130. Shah et al, G3, 2019 has been cited in line 32. Information about 6mA in human genome has been added to the table 1.
